# Identifying Suspect Bat Reservoirs of Emerging Infections

**DOI:** 10.3390/vaccines8020228

**Published:** 2020-05-17

**Authors:** Daniel Crowley, Daniel Becker, Alex Washburne, Raina Plowright

**Affiliations:** 1Department of Microbiology and Immunology, Montana State University, Bozeman, MT 59717, USA; alexander.washburn@montana.edu (A.W.); raina.plowright@montana.edu (R.P.); 2Department of Biology, Indiana University, Bloomington, IN 47405, USA; danbeck@iu.edu

**Keywords:** Nipah, Ebola, bats, phylofactor, phylogenetics

## Abstract

Bats host a number of pathogens that cause severe disease and onward transmission in humans and domestic animals. Some of these pathogens, including henipaviruses and filoviruses, are considered a concern for future pandemics. There has been substantial effort to identify these viruses in bats. However, the reservoir hosts for Ebola virus are still unknown and henipaviruses are largely uncharacterized across their distribution. Identifying reservoir species is critical in understanding the viral ecology within these hosts and the conditions that lead to spillover. We collated surveillance data to identify taxonomic patterns in prevalence and seroprevalence and to assess sampling efforts across species. We systematically collected data on filovirus and henipavirus detections and used a machine-learning algorithm, phylofactorization, in order to search the bat phylogeny for cladistic patterns in filovirus and henipavirus infection, accounting for sampling efforts. Across sampled bat species, evidence for filovirus infection was widely dispersed across the sampled phylogeny. We found major gaps in filovirus sampling in bats, especially in Western Hemisphere species. Evidence for henipavirus infection was clustered within the Pteropodidae; however, no other clades have been as intensely sampled. The major predictor of filovirus and henipavirus exposure or infection was sampling effort. Based on these results, we recommend expanding surveillance for these pathogens across the bat phylogenetic tree.

## 1. Introduction 

Filoviruses and henipaviruses are zoonotic viruses of public health concern. Spillover events have occurred in Asia (Nipah virus), Australia (Hendra virus), and sub-Saharan Africa (Ebola virus, Marburg virus, undescribed henipaviruses) [1,2,3,4]. Bats are reservoir hosts for most these viruses, and epidemiological studies have linked bats to incidence cases in spillover events [1,2,5,6]. 

There are unanswered questions regarding bat-virus interactions and the reservoir status of bat species, despite these public health concerns. For example, the biology of Marburg, Nipah, and Hendra viruses within bats is poorly understood, and the Zaire strain of Ebola virus has yet to be isolated from a bat [7,8]. While studies that have isolated virus have identified specific species as reservoirs, the phylogenetic breadth of suspected reservoirs has not been assessed. Understanding which species harbor these viruses is key for guiding surveillance in wildlife and understanding the pathways of pathogen spillover [9]. 

In order to assist surveillance decision making, we systematically collated data on sampling effort and detection of virus through PCR and detection of exposure through serology (hereafter “evidence of infection or exposure”) in bats for filoviruses and henipaviruses and searched for cladistic patterns while using a novel machine learning algorithm, phylofactorization [10]. Phylofactorization allowed us to flexibly identify patterns of virus hosting at any scale on the bat phylogenetic tree, not constraining our analysis to specific taxonomic scales, such as order or genus. To date, predictions of high-risk species have focused on trait-based analyses [11,12]. Importantly, traits determining reservoir host status may evolve along lineages; thus, any extraneous trait in that lineage might be identified as determining a natural host of a virus. Cladistics can circumvent such issues by identifying taxonomic groups more likely to contain reservoirs, and this information can be used to direct future trait-based analyses. By identifying taxonomic groups with a high propensity to be found with evidence of viral infection, clade-based analyses can identify high-risk lineages without needing to account for the phylogenetic dependence of traits or collinearity among trait-based predictors. We first used phylofactorization to identify bat lineages with a higher propensity to contain suspect reservoirs of filo—and henipaviruses while controlling for sampling effort. We then assessed how sampling effort aligned with evidence of infection across the bat phylogeny to guide resource reallocation for field surveillance.

## 2. Methods 

### 2.1. Dataset

We used a previously published dataset of prevalence and seroprevalence to assess taxonomic patterns in bat infection (PCR-based detection) and exposure (antibody-based detection) with filoviruses and henipaviruses in wild bats. Systematic searches were run in Web of Science, CAB Abstracts, and PubMed, as described previously [13]. Records with repeat measurements of different bodily substrates (i.e., RNA from urine and saliva from the same bats) were pooled while using a weighted average based on the number of hosts sampled. We interpret any PCR or serological evidence as evidence of current or prior viral infection or exposure. We used the count of “unique sampling events” for a virus in that bat species to account for sampling effort for each virus group. We defined a “unique sampling event” as an estimate of seroprevalence or prevalence in a specific species at specific time and place [13]. A study that sampled a species over a six month period, but did not specify the sampling dates and only reported a single seroprevalence estimate, would only count as a single sampling event.

### 2.2. Bat Phylogeny

We obtained a phylogeny of 1103 bat species from the Open Tree of Life while using the *rotl* package in R [14,15]. We used the *ape* package to prune the tree to bat species in our dataset and provide branch lengths using Grafen’s method [16,17]. We used the package *taxize* to obtain our taxonomy [18].

### 2.3. Phylofactorization of Bat Virus Data

We partitioned the bat phylogeny by sampling effort and evidence of infection while using phylofactorization [10,19] in the R package *phylofactor* (version 0.0.1, available at https://github.com/reptalex/phylofactor.). Phylofactorization partitions a phylogeny by iteratively identifying edges in a tree that maximize an objective function contrasting species separated by the edge. For regression phylofactorization of sampling effort, the objective function was the deviance of a categorical variable indicating which side of an edge each species is found. This categorical variable, the side of the edge which contained each species, was the explanatory variable in a negative binomial generalized linear model (GLM). The outcome variable in this GLM was the log(1+sampling effort). 

We computed the quantile of the deviance observed under the distribution of deviances obtained from 100 phylofactorizations of null datasets to assess statistical significance of phylofactored clades. Null data were simulated by randomly drawing sampling effort at each tip from a negative binomial distribution with mean and variance equal to that observed in our data. A statistically significant clade was one whose objective function was within the top 5% of the null distribution for that iteration of phylofactorization.

The phylofactorization of evidence of infection predicted a binary outcome variable: the presence or absence of evidence of viral infection or exposure, indicated by serology or PCR positivity. The objective function was the deviance of a categorical variable indicating which side of an edge each species is found. Bats that were not sampled for filoviruses or henipaviruses in the published literature were excluded from this analysis. We controlled for sampling effort by offsetting the predicted canonical parameter from binomial regression of sampling effort on evidence of viral infection or exposure. For improved resolution, the significance of phylogenetic factors was assessed while using 500 null phylofactorizations. The null distribution for evidence-of-infection was simulated as a vector of independent, identically distributed Bernoulli random variables with a probability equal to the fraction of viral-positive bats that were observed in our dataset.

## 3. Results 

### 3.1. Filovirus Suspect Reservoirs

When we ignored sampling effort, we identified two statistically significant clades for filovirus reservoirs: the Pteropodidae, which had a high proportion of species with evidence of infection or exposure, and the Rhinolophoidea, which had a low proportion of species with evidence of infection or exposure (Figure 1). In the superfamily Rhinolophoidea, only 8% of species (*n* = 39) had evidence of infection or exposure (*p* < 0.01 based on 500 null simulations). In the family Pteropodidae, 86% of species (*n* = 14) had evidence of infection or exposure (*p* < 0.01 based on 500 null simulations). Of the sampled species that were not contained in our phylofactored clades (the paraphyletic remainder), 35% (*n* = 79) had evidence of filovirus infection or exposure. Of the total 132 bat species that were sampled for a filovirus, 32.5% had evidence of infection or exposure.

After accounting for filovirus sampling effort, the Rhinolophoidea superfamily, for which only 8% of species had evidence of filovirus infection or exposure, remained statistically significant. The Pteropodidae family, for which 86% of species had evidence of filovirus infection or exposure, did not remain statistically significant after accounting for sampling effort (*p* > 0.05, based on 500 null simulations). The odds of the paraphyletic remainder having evidence of filovirus infection or exposure were 4.7 times the odds of the Rhinolophoidea bats having evidence of filovirus infection or exposure. Forty-three percent of the paraphyletic remainder (*n* = 89) had evidence of filovirus infection or exposure, when accounting for sampling effort (Figure 1).

When we analyzed the filovirus sampling effort data (the number of reported sampling events per species), we identified four statistically significant clades with different levels of sampling effort (Figure 2). The first clade was a sub-clade of the Pteropodidae family (*n* = 35, *p* < 0.01 based on 100 null simulations), consisting of Pteropodinae and Macroglossinae species; bats in this clade had an average of 5.7 unique sampling events. For context, the Pteropodinae were sampled 12.4 times as much as the species across our entire tree (*n* = 1087), where there were, on average, 0.46 sampling events per species. We next identified the superfamily Noctilionoidea (*n* = 176, *p* < 0.01 based on 100 null simulations), with zero reported sampling events across 176 species. The third clade was the genus *Hipposideros* (*n* = 37, *p* < 0.01 based on 100 null simulations), with an average of 1.38 unique sampling events. The final clade was the subfamily Molossinae (*n* = 36, *p* < 0.01 based on 100 null simulations), with an average of 1.28 unique sampling events per species. The paraphyletic remainder (*n* = 803) had, on average, 0.25 unique sampling events per species. Phylofactorization identified additional clades; however, a visualization of the scree plot of our objective function led us to exclude these clades from our results.

### 3.2. Henipavirus Suspect Reservoirs 

Without accounting for the sampling effort, two clades had a significantly higher propensity for evidence of henipavirus infection or exposure (Figure 1): the subfamily Pteropodinae (*n* = 14, 93% with evidence of infection or exposure, *p* < 0.01 based on 500 null simulations) and the genus *Rousettus* (*n* = 4, 100% with evidence of infection or exposure, *p* < 0.01 based on 500 null simulations). Only 14% of the paraphyletic remainder (*n* = 119) displayed evidence of henipavirus infection or exposure. When accounting for sampling effort, no bat clade had a statistically significant higher or lower propensity to have evidence of henipavirus infection (*p* > 0.05). The first identified clade, though non-significant, was the *Rousettus* genus, with 100% displaying evidence of infection or exposure to a henipavirus (*p* = 0.23, based on 500 null simulations). 

When we analyzed the henipavirus sampling effort data, we identified two statistically significant clades with different levels of sampling effort (Figure 2). The first clade identified was the Pteropodidae (*n* = 183, *p* < 0.01 based on 100 null simulations), which was well-sampled, with 2.26 unique sampling events per species. For context, the Pteropodidae were sampled 4.27 times as much as the species across our entire tree (*n* = 1087), where there were, on average, 0.53 sampling events per species. The final clade was the *Rhinolophus* genus (*n* = 52, *p* < 0.01 based on 100 null simulations), with 0.69 unique sampling events per species. The 852 species in paraphyletic remainder had an average of 0.15 unique sampling events per species.

## 4. Discussion

Despite the public health concerns surrounding filovirus and henipavirus spillover events, there has been little systematic effort to identify phylogenetic patterns in suspected bat reservoirs. Moreover, the sampling effort has not been assessed across species. Our analysis suggests that, after accounting for sampling effort, evidence of filovirus and henipavirus infection or exposure is widespread across the phylogenetic tree. We suggest that the extensive sampling efforts in the Pteropodidae (Old World fruit bats) could explain the commonly reported association of the Pteropodidae with these viruses. We recommend expanding surveillance of these viruses across the bat phylogenetic tree to better understand the phylogenetic distribution of these viruses. For filoviruses specifically, we recommend expanding surveillance to bat clades in the Western Hemisphere and performing more sampling in the Rhinolophoidea superfamily.

### 4.1. Filoviruses

Evidence of filovirus infection is widely distributed across the phylogeny of sampled bats, which suggests that these viruses are not constrained to a specific clade. The single exception to this was the Rhinolophoidea, which our analysis found had less evidence of filovirus infection or exposure. Interestingly, one of the six species that has been found with filovirus RNA is within the Rhinolophoidea clade (*Rhinolophus eloquens*) [20]. However, Marburg PCR-positive *R. eloquens* bats were co-roosting with Marburg PCR-positive *Rousettus aegyptiacus*. It is possible that these *R. eloquens* were infected by spillover from this co-habitating species.

While we found evidence of filovirus infection throughout the phylogeny, sampling has been concentrated in specific clades, notably the Pteropodidae, perhaps because early laboratory and epidemiological evidence linked filoviruses with this clade [5,21,22,23,24,25,26]. By contrast, there is a dearth of sampling in clades such as the Noctilionoidea. Leendertz et al. [7] came to a similar conclusion from a visual inspection of their data. Our results substantiate their conclusions and provide information regarding sampling efforts within and between clades. Specifically, we did not find that the Yangochiroptera, the parent clade of Noctilionoidea, were under-sampled for Ebola and Marburg virus, as the Molossinae have been sampled for filoviruses at rates that are comparable to the rest of the phylogeny.

Geography explains some of these phylogenetic patterns in the sampling effort. The Noctilionoidea, for example, are a South American lineage. Until 2020, no bats, to our knowledge, had been sampled for filoviruses in the Western Hemisphere. We predict that South American species are capable of being infected by filoviruses based on the phylogenetic breadth of evidence of infection among sampled bats. Han et al. (2016) arrived at a similar conclusion, using a trait-based analysis to predict that novel carriers of filoviruses are in South America. Our analysis, which was based on taxonomy alone, adds further weight to this conclusion. However, at the time of submission, there was a report of antibodies that cross-reacted with filovirus proteins in bats that were sampled in Trinidad [27]. This new analysis supports our conclusions, although the recency of this publication precluded it from our analysis.

Determining the age of the filovirus lineage could provide insights into the potential distribution of filoviruses in South America. There are two primary estimates for the age of the filovirus lineage: 10,400 years old [28] and ancient [29]. If the filovirus lineage is 10,400 years old, then New World bats may be biogeographically isolated from filoviruses yet competent reservoirs. In this case, viral introduction could be possible. More work is needed in order to substantiate the finding in Trinidad to determine whether filoviruses are widely distributed in the Western Hemisphere. Because our work suggests viral infection may be widespread across bat taxa, either filoviruses are likely widely distributed in the Western Hemisphere (but poorly sampled) or New World bats could be susceptible to the introduction of filoviruses from the Old World. We recommend that future research prioritize surveillance of filoviruses in the New World and experimentally investigate the capacity of Old-World filoviruses to infect New World bats.

Our filovirus results are in further dialogue with trait-based analyses of bat filovirus reservoirs. Han et al. [30] identified specific phenotypes associated with filovirus infection, notably adult and neonate body sizes and rates of reproductive fitness. Our analysis provides clade-specific likelihoods of suspect reservoir status in bats. We found strong phylogenetic signal in the filovirus data—specifically the Pteropodidae and Rhinolophoidea—suggesting that trait-based analyses should adjust for these clades and their shared traits as a step towards well-controlled phylogenetic comparisons. Failing to correct for this signal will identify the Pteropodidae or Rhinolophoidea traits as driving suspect reservoir likelihoods in bats. Consequently, phylofactorization serves as both an inferential tool for identifying lineages at-risk of hosting filoviruses and a stepping stone for phylogenetic comparative methods being applied to more complex trait-based analyses.

### 4.2. Henipaviruses

Pteropodinae species were more likely to have evidence of henipavirus infection than other bats; however, this association was no longer statistically significant when controlling for sampling effort. These results suggest that, while Pteropodinae remains important in henipavirus circulation and spillover, the scarcity of henipaviruses outside of Pteropodinae might be due to sampling effort rather than biological constraint. Unfortunately, sampling is sparse outside of the Pteropodinae and *Rhinolophus* bats. We calculated that nearly 80% of all unique henipavirus sampling events occurred within these two clades. The limited sampling outside of the Pteropodinae and *Rhinolophus* clades limits the comparisons we can make concerning phylogenetic patterns across the entire tree.

While our analysis suggests henipavirus reservoirs may not be confined to the Pteropodidae clade, sampling in these bats is certainly warranted, as Pteropodidae species are confirmed reservoirs for several henipaviruses through viral isolation [31,32], experimental infection (e.g., Nipah virus, Hendra virus) [33], and epidemiological links with spillover [34,35]. The concentration of evidence of henipavirus reservoir status in Pteropodidae is reflected in the cladistic patterns that were detected in sampling effort, with Pteropodidae being sampled more frequently when compared to the rest of the phylogeny. However, this current emphasis on hosts that are the likely source of spillover might miss other maintenance reservoir hosts.

### 4.3. Bombali Virus

The recent detection of a new ebolavirus, Bombali virus, in two insectivorous bat species, supports our finding that wider sampling, beyond Pteropodidae bats, will be necessary in order to understand the circulation of ebolaviruses in nature. With the exception of the Rhinolophoidea bats, where filovirus exposure has rarely been detected, our results suggest the main predictor of filovirus evidence of infection is sampling effort. The Bombali virus was detected in *Chaerephon pumilus* and *Mops condylurus*, two species in the Molossinae subfamily [36]. Phylofactorization identified the Molossinae as one of the most heavily sampled clades for filoviruses. The identification of a new filovirus in the Molossinae is not unexpected based on our analysis of the data.

### 4.4. Limitations

Our analysis was heavily reliant on serological data due to the scarcity of PCR evidence in the literature. The limitations of serology in wildlife disease diagnostics have been documented previously [37,38] and include cross-reactivity and a lack of specificity. For example, a reported positive Ebola virus result might represent an individual with antibodies that were generated in response to Ebola virus or to a virus with some antigenic similarity. However, virus infection is often at low prevalence, and it is variable across space and time, even in species that are known reservoirs [13,35]. Therefore, negative PCR results are difficult to interpret unless sampling efforts are intense with longitudinal and spatial coverage. The majority of studies we reviewed were cross-sectional, with small sample sizes [13]. In contrast to viral RNA, antibodies persist after the infectious agent clears, lengthening the window over which evidence of infection can be detected, and increasing the probability of detection [39]. Thus, serological data are useful in comparative analyses such as these, which are focused on determining clades that, despite being under-sampled, show evidence of prior or current viral infection or exposure.

As a caution, antibody cross-reactivity limits our ability to make inferences regarding specific viruses from positive serology. Thus, we limited our analyses and interpretations to coarse viral taxonomies rather than viral species. We did not use viral species-stratified data in our analysis, as serological measures taken from the literature could be measuring previously uncharacterized, but antigenically similar, viruses.

### 4.5. Evidence of Infection Metric

We used a binary outcome for evidence of infection, which improved the comparability of results across publications. A quantitative variable, such as counts of positive individual bats, would necessitate combining estimates across studies. However, methodological inconsistencies, such as different positivity cutoffs and different proteins used in diagnostics, are often not reported and their omission makes comparisons difficult [13]. Therefore, our use of a binary outcome collapsed important information contained in the data while also minimizing errors that limited more quantitative comparisons across studies.

### 4.6. Sampling Effort Metric

Our measure of sampling effort was the number of unique sampling events. Our sampling effort metric biased our assessment of suspected reservoir status away from heavily sampled clades, an example being the Pteropodidae. Our analysis only provides limited insight into whether the Pteropodidae have more evidence of viral infection due to heavy sampling or because they are more susceptible to infection and, thus, frequent subjects of study. Our analytical strategy provides stronger insight into under sampled clades that have evidence of infection, despite limited sampling and heavily sampled clades that have little evidence of infection. Our analytic strategy also highlights the importance of accounting for sampling effort in research moving forward, as our results did change when we accounted for our sampling effort metric.

### 4.7. Choice of Phylogenetic Methods

We used phylofactorization over alternative techniques, such as phylogenetic generalized least squares (PGLS) or phylogenetic generalized linear mixed models (PGLMMs) [16]. Such methods depend upon a Brownian motion model of evolution; however, previous work has suggested that punctuated equilibrium models of evolution may be more appropriate and these models are better analyzed by phylofactorization [10,40,41]. Furthermore, PGLS and PGLMMs are useful for controlling for phylogenetic dependence when identifying covariates that predict species-level outcomes [16]. Phylofactorization, unlike PGLS and PGLMMs, works by identifying cladistic patterns in the phylogenetic tree in a flexible manner while accounting for confounding variables [10]. Such cladistic patterns meaningfully simplify the classification of reservoirs through coarse-grained inferences about which lineages have unusually high or low odds of being a reservoir, having controlled for confounding variables, such as sampling effort [42].

### 4.8. Physiological Differences between Bat Clades

Our analysis is intended to identify patterns in virus infection or exposure across the phylogenetic tree. Unobserved traits that determine host compatibility with viruses, such as cell receptors, which enable or prevent viral entry, have likely evolved along specific clades. If these trait data existed in a comparable manner across species, they might explain the patterns that we observed in the Rhinolophoidea and filoviruses. We hope our analysis will help to drive new questions and the data collection required to explain the biological mechanisms behind the patterns that we observed.

## 5. Conclusions

Fruit bats are the presumptive reservoirs for henipaviruses and filoviruses. However, our analysis suggests that the association of Pteropodidae bats with these viral clades is driven primarily by sampling effort. We suggest expanding surveillance to under sampled clades to further elucidate the phylogenetic patterns in filovirus and henipavirus reservoirs. For filoviruses specifically, New World bats require more sampling in order to understand the distribution of filovirus reservoirs across the bat phylogeny.

## Figures and Tables

**Figure 1 vaccines-08-00228-f001:**
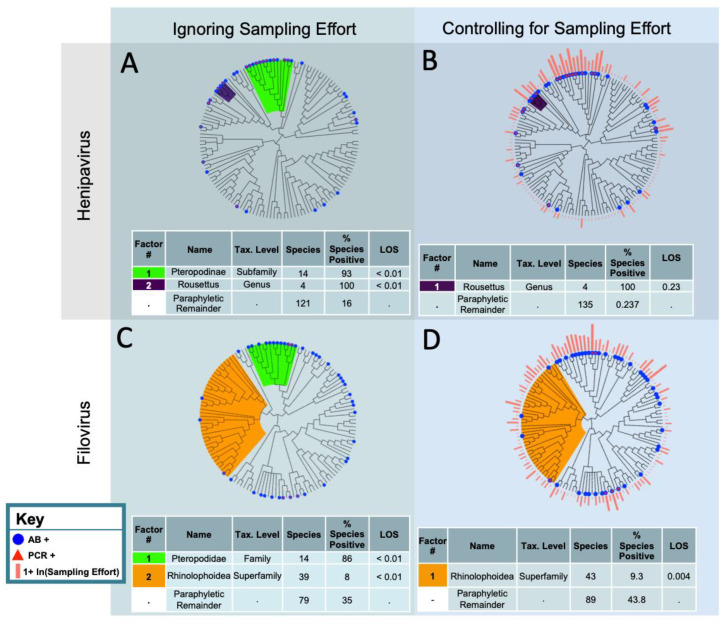
Phylofactorization of the bat species that have been PCR or antibody positive for filoviruses and henipaviruses. The phylogeny is from the Open Tree of Life. The analysis was performed using generalized phylofactor regression. The outcome variable is evidence of infection, determined using either PCR or antibody positivity. The factor name and taxonomic level was determined by identifying the most basal taxonomic grouping shared by all species in the clade. The % species positive is the proportion of species PCR or antibody positive within the factor. Level of significance (LOS) refers to the value of the objective function for a clade in relation to the distribution of objective function values from the null simulations. Factors are ordered from top to bottom by their LOS. (**A**) Analysis excludes bats that have not been sampled for henipaviruses and does not account for sampling effort. (**B**) Analysis excludes bats that have not been sampled for henipaviruses and does account for sampling effort. (**C**) Analysis excludes bats that have not been sampled for filoviruses and does not account for sampling effort. (**D**) Analysis excludes bats that have not been sampled for filoviruses and does account for sampling effort.

**Figure 2 vaccines-08-00228-f002:**
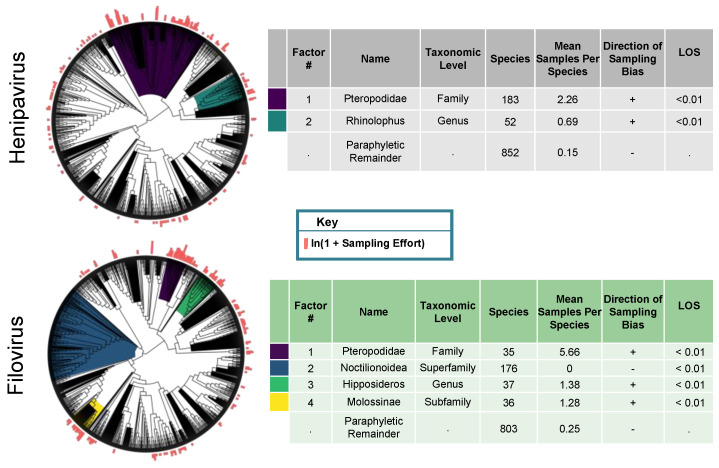
Phylofactorization of filovirus and henipavirus sampling effort across the full bat phylogeny. The phylogeny is from the Open Tree of Life. The outcome variable is the natural log transform of our metric of sampling effort. Our metric of sampling effort was the number of unique sampling events per species. The factor name and taxonomic level was determined by identifying the most basal taxonomic grouping shared by all species in the clade. Level of significance (LOS) refers to the value of the objective function for a clade in relation to the distribution of objective function values from the null simulations. Factors are ordered from top to bottom by their LOS. The taxonomic level was determined by identifying the most basal taxonomic grouping shared by all species in the clade.

## Data Availability

The datasets generated and analyzed during the current study will be made available in a repository hosted at https://github.com/BozemanDiseaseLab.

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
