# Peer review of "Identifying Suspect Bat Reservoirs of Emerging Infections"

_vaccines, 2020, doi:10.3390/vaccines8020228_

Round 1
Reviewer 1 Report
It has long been recognized that bats constitute a major reservoir for several pathogenic viruses with filoviruses and henipaviruses posing the greatest risk of causing a pandemic in the human population. In this manuscript, the authors present an analysis of the distribution of members of these two virus families among several families in the bat population.
Soon after the two major pathogenic henipaviruses , Nipah and Hendra, emerged in Asia and Australia, respectively, it was recognized that fruit bats were the natural reservoir for both viruses. However, this manuscript clearly establishes that these viruses are likely not limited to this particular bat species. Indeed, from the data presented here, it is apparent that several other species, most alarmingly those in the Western hemisphere, including South American species, are quite capable of also being infected by, as well as serving as natural reservoirs for, these viruses. This, of course, raises the possibility of the high mortality diseases caused by these viruses emerging in the Western hemisphere.
Corresponding findings ae also presented for the filovirus family. While these viruses, ebola and Marburg, have been detected in two bat species, studies described here indicate that a third of the total species are actually infected. The assignment of the Pteropodidae as the carrier of these viruses derived basically from the focus almost exclusively on this species driven by early laboratory and epidemiological evidence linking ebola to them.
However, based on a novel algorithm called phylofactorization used to survey the bat phylogeny for patterns of henipavirus and filovirus infection, the point made very convincingly here is that our present understanding of the spread of these families across the bat population is profoundly biased by our limited sampling effort. To coin a phrase, we have been “looking under the lamppost”, i.e. .sampling only those bat species native to the area of the virus’ emergence and the disease outbreak.
This is an outstanding manuscript that makes a very strong case for an expansion of our sampling efforts for each of these viruses to include bat species encompassing as many species as possible, especially those endemic to the Western hemisphere. The timeliness of the manuscript couldn’t be more evident in light of the current Coronavirus pandemic engulfing the world. Indeed, I would be surprised if these studies were not soon expanded to include these viruses.
One caveat that I feel has not been addressed here is exactly how these different bat families differ from each other. In other words, is there anything about the members of each family in a physiological sense that could reasonably prevent them from harboring any of these viruses? I would suspect not. If so, this point should be made at some point in the manuscript. Other than this point, the manuscript makes an important contribution to our understanding of the bat reservoirs of pathogenic viruses and presents a real challenge that, if addressed, would help prepare us for future pandemics.
Author Response
Reviewer 1 asked us about physiological differences between bat clades and how this might impact the distribution of viral reservoirs across the tree. We agree this is one of the critical questions underlying the paper. Our analysis can identify patterns, but without the requisite data we can only speculate why these patterns exist. Ideally, we want this paper to drive research which formulates hypotheses and collects the data which address these patterns. I've included another section, starting on line 324, which I hope addresses the reviewer's important point and presents our thoughts on the issue:
Our analysis is intended to identify patterns in the phylogenetic tree. Unobserved traits that determine host compatibility with viruses, such as cell receptors which enable or prevent viral entry, likely have evolved along specific clades. If these trait data existed they might explain the patterns we observed in the Rhinolophoidea and filoviruses, for example. We hope our analysis will help drive new questions and the data collection required to explain the biological mechanisms behind the patterns we observed.
Reviewer 2 Report
The authors present an interesting study that uses the machine-learning algorithm, phylofactorization, to explore the relationships between the phylogeny of bat species, potential biases in sampling, and how these might be related to the identification of the host reservoir species for zoonotic filoviruses and henipaviruses. Overall the study concludes that, the sampling effort was the predictor for determining exposure to filoviruses and henipaviruses for bat species. The study highlights the issue that if we are to fully understand the zoonotic potential of these important virus families (and others presumably) and the role that bats play (or might play), future studies are required which systematically sample representative species across the bat phylogenetic tree. The manuscript will be of interest to those working in the field of zoonotic diseases and more broadly to understand how to determine the risk bats may pose in the context of current and future pandemics.
I have made some suggestions for the authors to consider:
Line 10 suggest revision:
Bats host a number of potential pathogens that following onward transmission to humans and/or animals can result in severe or fatal disease.
The current sentence implies that the pathogens are causing severe disease in bats, which is somewhat at odds with them being a reservoir host. In addition, one of the proposed pathways of humans contracting Ebola virus is through the contact/consumption infected “bush” meat, rather directly from bats.
Line 64-65 Suggest reorganisation of sentence:
“Any PCR or serological evidence we conclude is evidence of current or prior and or viral exposure.”
Lines 257-265 Please add some appropriate citations to support this statement of facts around Bombali virus.
Author Response
Reviewer 2 suggested we change the wording on line 10. We have changed the writing to clarify that the pathology occurs in humans and domestic animals, not in bats. We have also incorporated their suggestion on lines 64-65, which is now on lines 68-70. Last, we added appropriate citations on line 270 in regard to the Bombali virus.